# Integration of small RNA, degradome and proteome sequencing in *Oryza sativa* reveals a delayed senescence network in tetraploid rice seed

**Baosheng Huang[1,2], Lu Gan[1], Dongjie Chen[2], Yachun Zhang[1], Yujie Zhang[1], Xiangli Liu[1], Si Chen[1], Zhisong Wei[1], Liqi Tong[1], Zhaojian Song[1,3], Xianhua Zhang[1,3], Detian Cai[1,3], Changfeng Zhang[2]\*, Yuchi He[1,3]\***

**1** State Key Laboratory of Biocatalysis and Enzyme Engineering, School of Life Sciences, Hubei University, Wuhan, China, **2** Shandong Provincial Key Laboratory of Storage and Transportation Technology of Agricultural Products, Jinan, China, **3** Wuhan Polyploid Biology Technology Co. Ltd, Wuhan, China

\* hyc@hubu.edu.cn (YH); zcf202@163.com (CZ)

**Data Availability Statement:** All relevant data are within the paper and its Supporting Information files.

## Abstract

Seed of rice is an important strategic resource for ensuring the security of China's staple food. Seed deterioration as a result of senescence is a major problem during seed storage, which can cause major economic losses. Screening among accessions in rice germplasm resources for traits such as slow senescence and increased seed longevity during storage is, therefore, of great significance. However, studies on delayed senescence in rice have been based mostly on diploid rice seed to date. Despite better tolerance have been verified by the artificial aging treatment for polyploid rice seed, the delayed senescence properties and delayed senescence related regulatory mechanisms of polyploid rice seed are rarely reported, due to the lack of polyploid rice materials with high seed set. High-throughput sequencing was applied to systematically investigate variations in small RNAs, the degradome, and the proteome between tetraploid and diploid rice seeds. Degradome sequencing analysis of microRNAs showed that expression of miR-164d, which regulates genes encoding antioxidant enzymes, was changed significantly, resulting in decreased miRNA-mediated cleavage of target genes in tetraploid rice. Comparisons of the expression levels of small RNAs (sRNAs) in the tetraploid and diploid libraries revealed that 12 sRNAs changed significantly, consistent with the findings from degradome sequencing. Furthermore, proteomics also showed that antioxidant enzymes were up-regulated in tetraploid rice seeds, relative to diploids.

## Introduction

Rice (*Oryza sativa* L.) seed is the core of rice production, the "chip" of the rice industry. It is a strategic resource to ensure food and ecological security. Rice is consumed by more than half of the world's population [1]. The economic losses caused by seed aging are immeasurable. The delayed senescence ability of seed is associated directly with seed longevity. Therefore, further elucidation of the delayed senescence mechanism in seeds is essential, as is improvement

**Funding:** This project was supported by the Chinese National Natural Science Foundation (Grant Nos.31960068, 31270356, 31271690, and 31571639), 2017 Hubei Science and Technology Department Innovation Team (2017CFA023), 2016 Wuhan Yellow Crane Talents (science) Foundation. The funders had no role in study design, data collection and analysis, decision to publish, or preparation of the manuscript. Wuhan Polyploid Biology Technology Co. Ltd provided support for this study in the form of salaries for ZS, XZ, DC, and YH. The specific roles of these authors are articulated in the "author contributions" section. The funders had no role in study design, data collection and analysis, decision to publish, or preparation of the manuscript.

**Competing interests:** The authors have read the journal's policy and the authors of this manuscript have the following competing interests: ZS, XZ, DC, and YH are paid employees of Wuhan Polyploid Biology Technology Co. Ltd. This does not alter our adherence to PLOS ONE policies on sharing data and materials. There are no patents, products in development or marketed products associated with this research to declare.

of the delayed senescence ability of rice seeds, both playing a key role in the rice seed industry and agriculture. Seed vigor is an important, comprehensive index of seed quality, which is evaluated in terms of seed germination, seedling growth potential, etc. Seed vigor reaches its highest level during the physiological maturity period, from when it begins to decline irreversibly. This irreversible change is due to the process of seed senescence [2]. The use of artificial aging methods has become a common method to investigate the mechanisms of seed senescence, the influences of seed vigor, RNA integrity, DNA repair, etc. [3, 4]. The causes of seed senescence are very complex, and include environmental factors, free radical damage, mitochondrial aging, lipase abnormalities, etc. [5–8]. The dynamic changes and regulatory mechanisms of the rice seed senescence process have been widely researched. In the process of seed senescence, mitochondria are damaged firstly, resulting in considerable accumulation of reactive oxygen species (ROS). The role of mitochondria aging is a hot topic in seed senescence research [2]. The phenomenon of seed senescence being associated with mitochondrial damage has been confirmed in rice, maize, *Arabidopsis*, and other plants [9–11]. Wang reported that high humidity significantly reduced the seed lifespan of 'Huanghuazhan' rice [12]. Dong found that the quantitative trait locus (QTL) gene *qSS6-1* was related to storage tolerance of seeds [4]. Recent studies have shown that the microRNAs miR164c and miR168a can also regulate rice seed vigor [13]. The ability to tolerate senescence directly affects rice seed longevity. Previous studies showed that polyploid rice had greater stress tolerance potential than diploid rice, and also showed significant advantages in terms of drought and salt tolerance [14, 15]. However, research into rice delayed senescence is currently mostly based on diploid rice seed. In spite of the greater tolerance potential exhibited by polyploid rice seed, the delayed senescence properties and associated regulatory mechanisms in polyploid rice seed are rarely reported.

Polyploidy is an important factor in plant evolution and has been used in the development of new varieties in some crops. The increase in the ploidy level of polyploid rice, relative to diploid rice, can not only increase the copy number of tolerance genes, increasing the range of genetic variation, but can also increase the probability of achieving good gene combinations or producing fertile hybrids from wide interspecific crosses, thus enhancing the adaptability of rice to extreme environments [16, 17]. Polyploid rice breeding is a new way of breeding, which has the potential to greatly increase rice yield, but the seed-setting rate of most polyploid rice lines is less than 50% [18, 19]. Due to the lack of high-yielding polyploid rice accessions, there have been few reports on delayed senescence characteristics of polyploid rice seeds. Polyploidy rice has greater stress resistance and tolerance potential than diploid rice and has obvious advantages in terms of drought tolerance, salt tolerance, and seed longevity [14, 15]. Polyploid meiosis stability (PMeS) rice, with high seed set, has the double advantages of high yield and stress tolerance, and is a uniquely valuable material on which to carry out stress physiology research, including research into seed longevity [14, 15, 19]. The successful breeding of PMeS has broken through the difficulty of low seed setting, which has hindered the research into polyploid rice for decades. The hypothesis of gene balance holds that there is a change in gene copy number ("gene dose") in polyploids, although the abundance of gene products is not simply doubled or decreased. After polyploidization, interactions between genes form a complex control network through gene dose balance, to maintain the dose balance between the proteins encoded [20]. Artificial aging experiments on tetraploid and diploid rice seeds showed that the germination rate, seedling growth rate, and antioxidant enzyme activity of tetraploid rice were all significantly higher than those of their diploid rice parent, with polyploid rice having better delayed senescence potential than diploid rice. Therefore, these near-isogenic diploid/tetraploid lines are of great significance for analysis of the gene regulatory network involved in the delayed senescence mechanism operating in tetraploid rice seed.

As with other crops, seed senescence is one of the most important factors affecting rice growth and yield. The control of low temperature, low humidity and good ventilation conditions can ensure the safe storage of grain crop seeds. In order to satisfy the above-mentioned conditions, we need to build some high standard seed storage facilities, which will consume a lot of energy or resources. Improving the seed storability may be a cost-effective strategy. Therefore, the creation of high-yielding storage-tolerant rice varieties is of great significance for improving national food security and food safety in China and other countries dependent on rice as a staple food crop. Recently, with the development of rice genomics research, the functions of many genes have been identified experimentally [21]. It has been shown that plant stress tolerance can be regulated by oxidative stress and DNA/RNA damage [22]. RNA is easily degraded by ROS damage. A change in mRNA steady-state level may be caused by transcriptional regulation of mRNA degradation, and it has been confirmed that mRNA stability is directly related to stress tolerance [23]. With the destruction of RNA integrity, the function of RNA transport and signal transduction will be lost, and the ability of RNA to regulate protein synthesis will also decrease. The key reason for the decrease in seed germination rate is the limited expression of some genes related to germination [24–27]. Margaret et al. studied the seed vigor of soybean seeds which were stored dry at 5°C for 27 years, the results showing that the RNA integrity of seeds decreased gradually with increased duration of storage time, and showed a significant positive correlation with seed aging tolerance [28].

MicroRNAs (miRNAs) are 20 to 24 nt noncoding RNAs that regulate transcript expression by targeting mRNAs for cleavage or translational repression [29]. In plants, primary miRNAs are transcribed by RNA polymerase II and then processed by Dicer-like (DCL) proteins into precursors (pre-miRNA) with stem-loop structures [30]. The mature miRNAs are incorporated into the RNA-induced silencing complex (RISC) to target specific mRNAs and hence to down-regulate the expression of the target mRNAs [31]. Increasing evidence indicates that miRNAs have an influential role in numerous processes in plants, including development, abiotic stress tolerance, nutrient starvation response, and metabolism [32]. Previous studies indicated that seeds of different rice genotypes differ in their gene expression profiles during the aging process.

Integrating high-throughput sequencing and quantitative proteomics analyses will promote a more complete understanding of the molecular mechanisms underlying the involvement of specific genes in tetraploid and diploid rice seeds during the aging process. A pair of near-isogenic tetraploid and diploid rice accessions were used to study potential delayed senescence miRNAs and their target genes in rice seeds. Two libraries of small RNAs (sRNAs) were constructed from tetraploid or diploid rice, and then sequenced using the Illumina sequencing platform. Degradome sequencing was applied to directly detect cleaved miRNA targets at a global transcriptome level in diploid and tetraploid rice. The joint analysis of the proteomic, the miRNA and the degradome sequencing results may provide a more powerful explanation for a delayed senescence network in tetraploid rice seed. Analysis of molecular function regulation at the level of sRNAs, target genes, and proteins, may provide the foundation for understanding the delayed senescence and storage response mechanisms in tetraploid, relative to diploid, rice seed.

## Materials and methods

### Rice seed

Seed of *indica* rice (*Oryza sativa* ssp. *indica*) accessions 9311-2x and 9311-4x were selected as the experimental materials. Accession 9311-2x is a conventional diploid rice, whereas 9311-4x is a derivative of 9311-2x, using tissue culture and colchicine induction to induce chromosome

doubling to form tetraploid rice. In November 2016, field planting was carried out at the South breeding base of Hubei University in Lingshui County, Hainan Province, China. The seed from each accession was harvested and collected in May 2017, stored in a -20˚C refrigerator, following outdoor drying.

## Artificial aging treatment of rice seed and assessment of seed vigor

The artificial aging treatment parameters were 45˚C, 85% RH for an aging period of 24 d in dark. Seed of indica rice (Oryza sativa ssp. indica) accessions 9311-2x and 9311-4x. Each accession of 3000 seeds were artificially aged. 300 seeds were taken out from each group every 96 hours. 100 seeds were put into the biochemical incubator (6000 lx of light, 30˚C) for quantification of each germination variable, namely germination rate, germination index, growth potential and vigor index. Twenty seeds for antioxidant enzyme activity assessment. The remaining seeds were put into—80˚C refrigerator for molecular biology research.

Germination rate (%) = number of seeds normally germinated × 100/ number of seeds tested

Germination index (GI) = ∑GT/DT, where: GT refers to the number of days to germination after seed soaking; DT refers to the corresponding number of days to germination;

To measure seedling growth potential, the seeds were germinated and grown in the biochemical incubator for 7 d, and the length of the first leaf was measured

Vigor index (VI) = s × GI, where: s stands for seedling growth (length or weight).

## RNA extraction and assessment of RNA integrity

Six samples of rice seeds (three biological replicates of each of rice accessions 9311-2x and 9311-4x) were ground with liquid nitrogen. After grinding, total RNA was isolated and purified from each sample, using TRIzol reagent (Invitrogen, Carlsbad, CA, USA), following the manufacturer's procedure. The amount and purity of the RNA from each sample was quantified using a NanoDrop ND-1000 spectrophotometer (NanoDrop, Wilmington, DE, USA). The RNA integrity was assessed using an Agilent 2100 Bioanalyzer System (California, USA). To assess rRNA integrity, 1 μg RNA was electrophoresed on 1.5% agarose gel, stained with ethidium bromide [33], and visualized, and the intensities of the 28S and 18S rRNA bands were quantified after image analysis.

## Degradome and small RNA library construction and sequencing

Poly(A) RNA was purified from total plant mRNA (20 μg), using poly-T oligo-attached magnetic beads using two rounds of purification. Because the 3′ cleavage product of the mRNA contains a 5′-monophosphate, the 5' adapters were ligated to the 5' end of the 3′ cleavage product of the mRNA by the RNA ligase. The next step was reverse transcription to make the first strand of cDNA with a 3′-adapter random primer, and size selection was performed with AMPure XP beads (Beckman, USA). Then, the cDNA was amplified by PCR under the following conditions: initial denaturation at 95˚C for 3 min, 15 cycles of denaturation at 98˚C for 15 s, annealing at 60˚C for 15 s, and extension at 72˚C for 30 s, before a final extension at 72˚C for 5 min. The average insert size for the final cDNA library was 200−400 bp. Finally, we performed the 50-bp single-end sequencing on an Illumina HiSeq 2500 (LC Bio, China) following the vendor's recommended protocol. Small RNA (sRNA) libraries were constructed from the six samples. The sRNAs ranged in size from 18 nt to 30 nt and were purified from total RNA and ligated to 5′ and 3′ RNA adapters. These RNAs were reverse-transcribed into cDNAs and then amplified by PCR to obtain sufficient product for sequencing. Finally, the PCR products were subjected to Illumina sequencing at Lianchuan Co., Ltd., Hangzhou, China.

## Protein isolation, proteolysis, and TMT labeling

Rice seeds were ground into powder in liquid nitrogen, using a lysis buffer (Roche). The resulting samples were then ultrasonically disrupted for the extraction of total protein. After centrifugation at $10,000 \times g$ for 30 min at 4˚C, the supernatant from each extract was collected, and protein concentrations were determined using an enhanced BCA (bicinchoninic acid) Protein Assay Kit (P0010; Beyotime Biotechnologies, Ltd., Beijing, China), according to the manufacturer's instructions. Each protein sample (200 μg) was mixed with DL-dithiothreitol and alkylated with iodoacetamide, and then digested with trypsin overnight at a trypsin-to-protein ratio of 1:100. After trypsin digestion, the peptide mixture was desalted by elution from a Strata-X C18 SPE column (Phenomenex, USA) and vacuum dried. The peptides were reconstituted in 0.5 M triethylammonium bicarbonate (TEAB) buffer and processed according to the manufacturer's protocol for the Tandem Mass Tag (TMT) kit. Briefly, one unit of TMT reagent was thawed and reconstituted in acetonitrile (ACN). The peptide mixture was then incubated with the TMT reagent for 2 h at room temperature and pooled, desalted on a C18 SPE column, and dried by vacuum centrifugation.

## Liquid Chromatography-Tandem Mass Spectrometry (LC- MS/MS) analysis

Eluted fractions were lyophilized using a centrifugal speed vacuum concentrator (CentriVap® Complete Vacuum Concentrator; Labconco, Kansas City, MO, USA) and dissolved in 0.1% formic acid. Equivalent amounts of peptides from each fraction were mixed and then subjected to reversed-phase nanoflow LC-MS/MS analysis using a high-performance liquid chromatography system (EASY-nLC™; Thermo Fisher Scientific, USA), connected to a hybrid quadrupole/time-of-flight mass spectrometer equipped with a nano-electrospray ion source. The peptides were separated on a C18 analytical reverse-phase column with gradients of Solution A (0.1% formic acid in water) and Solution B (0.1% formic acid in ACN). A full MS scan was conducted using a Q Exactive™ mass spectrometer (Thermo Fisher Scientific, USA).

## Database search and protein identification and quantification

For peptide identification and quantification, MS/MS data were searched for against the assembly file, using the Mascot 2.2 and Proteome Discoverer™ 1.4 software (Thermo Fisher Scientific, USA). A unique protein, with at least two unique peptides that had a false discovery rate (FDR) <0.0160, was used for data analysis. Protein quantification was based on the total intensity of the assigned peptides. The average of eight labeled sample mixes was used as a reference and was based on the weighted average of the intensity of reported ions in each peptide identified. The final protein ratios were normalized to the median average protein content of the 8plex samples. Fold change values (FC) > 1.2 for upregulated or FC < 0.83 for downregulated proteins were set as the threshold for identifying differentially expressed proteins.

## Bioinformatics analysis

After the removal of low-quality tags, 3′ adapters and adapter-adapter ligation products, clean tags ranging from 18 to 30 nt were obtained. These sRNAs were annotated into different classes such as rRNA, tRNA, small nuclear RNA (snRNA) and small nucleolar RNA (snoRNA), as mapped to Rfam (http://rfam.sanger.ac.uk/) and NCBI (http://www.ncbi.nlm.nih.gov/) databases. Known microRNAs (miRNAs) were searched for by comparing sRNA tags with the miRbase 20 database with a nearly perfect match (mismatch ≤ 2). The remaining sequences were mapped to the genome of rice (*O. sativa*) with a perfect match to obtain their precursor

sequences. To identify the putative precursors, ~450-nt fragments were extracted by extending 200 nt from the two ends of the sRNA mapping loci, respectively. The secondary structure was predicted by the MIREAP (https://www.mireap.net/) program and the minimum free energy was set below -20 kcal mol$^{-1}$ as required. In addition, the miRNA and miRNA pairs should satisfy the requirement of the 2-nt overhangs on the 3′ end.

### Determination of antioxidant enzyme activities in rice seeds

Superoxide dismutase (SOD) activity was determined by the spectrophotometric nitroblue tetrazolium (NBT) method. Peroxidase (POD) activity was determined by the guaiacol method and catalase (CAT) activity was determined by UV spectrophotometry [34].

Extraction of enzyme solution: Take 0.1g rice seed, add 1ml 0.05mol/l phosphate buffer (pH = 7.8), then grind in ice bath and centrifugal treatment at 10000 rpm/min for 20 min, the supernatant should preserved at 4˚C for use.

Determination of superoxide dismutase (SOD) activity: Take 200μl enzyme solution, add 3ml 0.05mol/L phosphate buffer (pH 7.8), 600μl 130mmol/L methionine solution, 600μl 750μmol/L nitro-blue tetrazolium(NBT) solution, 600μl 100 μmol/L riboflavin solution, 600μl 100μmol/L EDTA-Na2 solution, and reacted for 20 min under 4000lx of light. The absorbance was measured at 560nm by spectrophotometer.

Determination of catalase (CAT) activity: Take 50 μl enzyme solution and 2ml 0.1mol/lphosphate buffer (pH = 7.0), 25˚C water bath for 5min, add 0.5ml 0.1mol/l H2O2, vortex mixed, The enzyme activity was measured at 240nm and read every 20s for 3 times. The absorbance change value per minute (ΔA240/min·g FW) was used to represent the enzyme activity.

Determination of peroxidase (POD) activity: Take 20 μl enzyme solution and 2ml 0.3% guaiacol, 30˚C water bath for 5min, 1ml of 0.3% H2O2 was added, and vortex mixed. The enzyme activity was measured at 470 nm and read every 20s for 3 times. The absorbance change value per minute (Δ A470/min·gFW) was used to represent the enzyme activity.

### Determination of malondialdehyde content in rice seeds

Malondialdehyde was extracted using trichloroacetic acid solution and reacted with thiobarbituric acid (TBA) to form a pink compound. The absorbance value at 532 nm was measured with a spectrophotometer and compared with the absorbances of a calibration series.

### Statistical analysis

Data processing was carried out by Excel, SPSS Statistics 25.0 (IBM, Armonk, NY, USA), and the R language package. Each experiment was repeated three times. The one-way analysis of variance or unpaired t-test was used to test statistical significance between groups. Data are presented as the least squares means ± standard error of the mean (SEM). Differences were considered significant at the $p < 0.05$ level.

## Results

### Assessment of response of 9311-2x and 9311-4x to artificial aging treatment

The germination experiments were carried out for seed of *indica* rice accessions 9311-2x and 9311-4x, subjected to artificial aging. Observations were carried out and recorded every 24 h. On the seventh day, indexes of germination rate, germination index, growth potential, and vigor index were determined (Fig 1). The effect of the two ploidy level rice accessions, with respect to vigor index, growth potential, and germination index over the duration of the

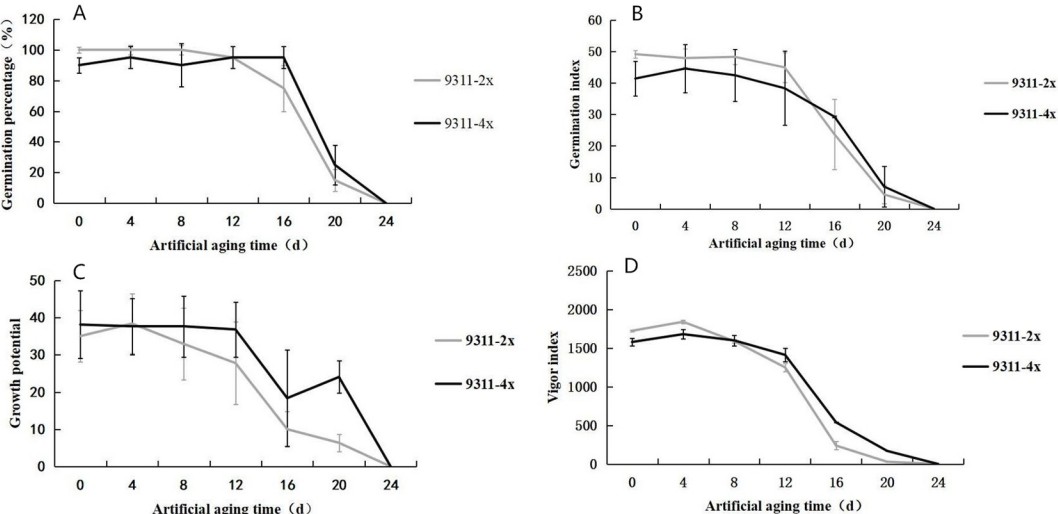

**Fig 1. Effect of artificial aging treatment on seed vigor for accessions 9311-2x and 9311-4x.** (A) The response of germination rate to the duration of the artificial aging treatment. (B) The response of germination index to the duration of the artificial aging treatment. (C) The response of growth potential to the duration of the artificial aging treatment. (D) The response of seed vigor to the duration of the artificial aging treatment. Values were mean ± SE.

artificial aging treatment, was basically the same as that of the germination rate. The germination ability of rice seeds decreased with the duration of aging. At 0 d, 4 d, 8 d, and 12 d, the germination percentage and the germination index of 9311-4x was lower than that of 9311-2x. Meanwhile, the vigor index of 9311-4x was lower than that of 9311-2x at 0 d and 4 d. After 24 days of artificial aging treatment, the germination rate of both accessions had decreased markedly. However, the values for 9311-4x were higher than those of 9311-2x after 16 d, 20 d, and 24 d. It was considered that seed of the tetraploid rice accession 9311-4x had greater storability than that of the diploid accession as the duration of aging increased.

## The RNA Integrity Number (RIN) of rice seed for accessions 9311-2x and 9311-4x during artificial seed aging treatment

During the process of the artificial aging treatment, the RIN was determined and it was found that, at the beginning of the artificial aging treatment, the RIN for 9311-4x seeds was 7.8, and the RNA peak area of 18S and 25S accounted for 35.1% (Fig 2A). After 12 d of artificial aging, the RIN had decreased to 6.6, and the proportion of the RNA peak area attributable to 18S and 25S had decreased to 20.6% (Fig 2B). On the other hand, the RIN of accession 9311-2x at the start of artificial aging was 6.7, and the area of 18S and 25S RNA peaks accounted for 21.9% (Fig 2C). After artificial aging for 12 days, the RIN had decreased to 6.6, and the proportion of RNA peak area of 18S and 25S decreased to 18.7% (Fig 2D). It was clear that, as aging progressed, the RIN was higher for 9311-4x than for 9311-2x, with artificial aging leading to a decrease in RIN, as well as an increase in sRNA peak area and in RNA degradation. RNA integrity was positively correlated with the germination rate of rice seeds, especially after 12 days of artificial aging, at which point the RIN decreased rapidly, and the germination rate also decreased rapidly. The rates of decrease of RIN and germination rate in rice seed 9311-4x were slower than those in 9311-2x (Fig 3). These data allowed the proposal of an experimental hypothesis that the RIN was positively correlated with seed vigor, with the seed vigor of 9311-4x being higher than that

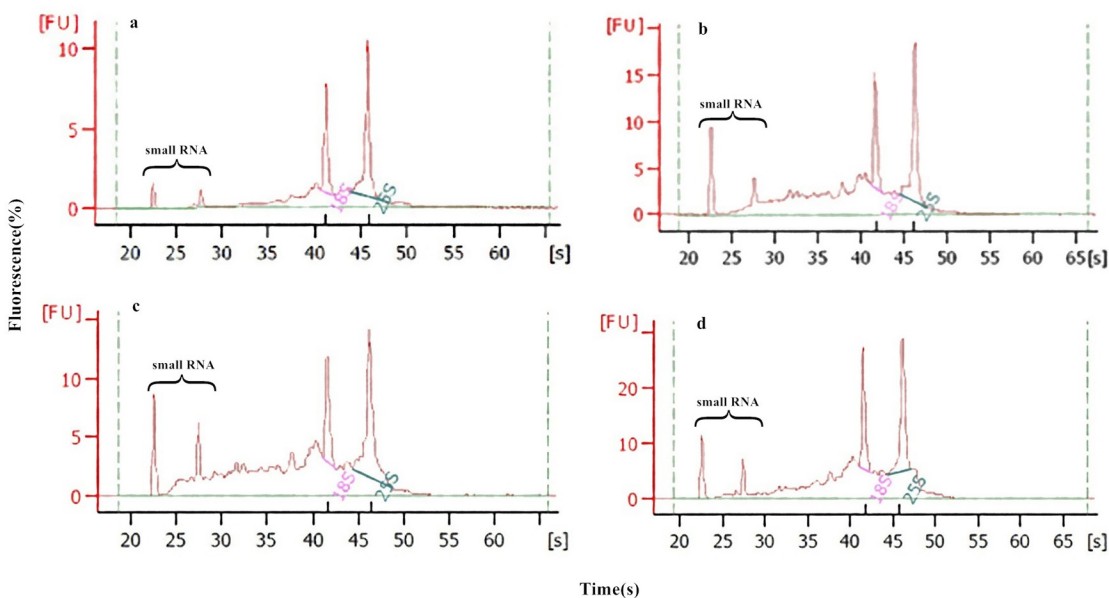

**Fig 2. The percentage of maximum RNA fluorescence (FU) during artificial aging for diploid and tetraploid rice seed.** (a) Rice seed of 9311-4x, at the beginning of artificial aging. (b) Rice seed of 9311-4x, after 12 days artificial aging. (c) Rice seed of 9311-2x, at the beginning of artificial aging. (d) Rice seed of 9311-4x, after 12 days artificial aging.

of 9311-2x. In the following experiment, we focused on RNA integrity by omics analysis, to investigate the relationship between seed vigor and RNA damage in rice seeds.

## Overview of sRNA sequencing results

To identify miRNAs from diploid and tetraploid rice seeds, two sRNA libraries were constructed. Raw sRNA sequencing reads have been deposited at NCBI (https://www.ebi.ac.uk/). After removing the low quality sequences, adapters, and sequences smaller than 16 nt or larger than 30 nt, 28,117,969 clean reads in 9311-2x and 30,297,197 clean reads in 9311-4x were obtained. Analysis of the length distribution of unique sRNA reads showed that the two libraries contained similar data, with the 21-nt sRNAs being the most abundant for both accessions, and this result was consistent with previous studies in *O. sativa* [35] (Fig 4).

We mapped the unique sRNAs onto miRBase 20 to identify the known miRNAs based on the criterion of mismatch ≤ 2. As a result, 40 known and 58 novel miRNAs were identified; the miRNA ID and the corresponding sequences were shown in S1 Table. miRNAs were mapped onto the genome sequences of *O. sativa* to discover their precursors and to determine their locations on specific rice chromosomes. As a result, 98 precursors were discovered, including 40 conserved miRNA precursors and 58 novel miRNA precursors (S2 Table).

To identify the miRNAs which were differentially expressed between 9311-2x and 9311-4x, the expression abundance of all the known miRNAs was compared using the *P*-value based on the filter parameter of fold change >1.2 and FDR for *P*-value <0.05, after being subjected to TPM normalization. As a result, a total of 12 miRNAs showed differential expression between 9311-4x and 9311-2x (Table 1). Of these, three miRNAs were downregulated in the tetraploid group, compared with the diploid group. Osa-miR164d, a miRNA which interact with the rice gene which encodes Cu-Zn superoxide dismutase, was down-regulated in 9311-4x (Table 1). MiR164d target four genes, including Os02g0579000, Os06g0675600, Os12g0610600, and

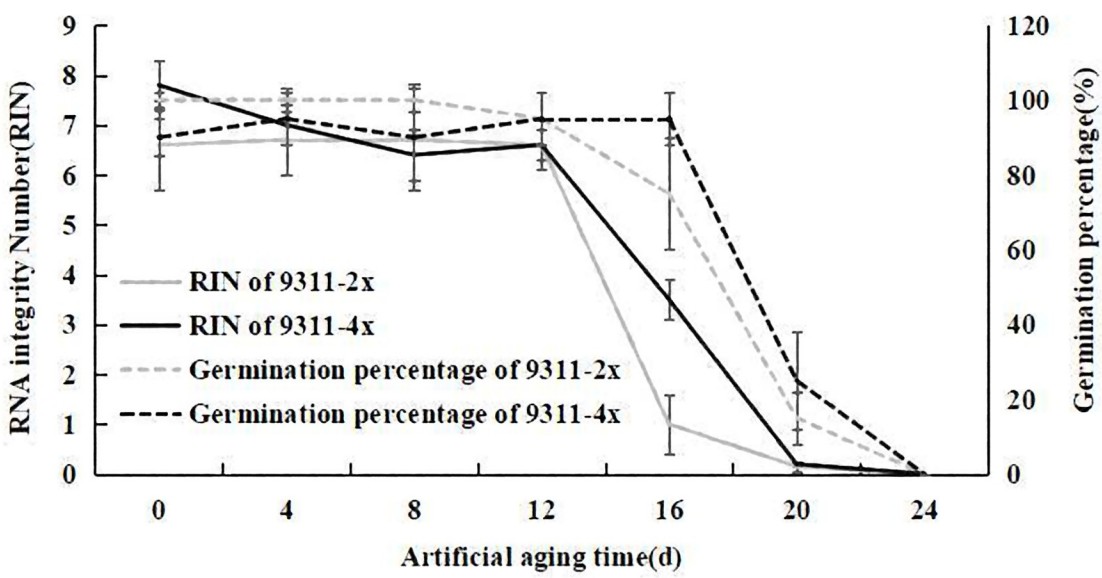

**Fig 3. RNA quality and germination rate of 9311-2x and 9311-4x seed aged for up to 24 days at 45˚C and 85% RH.** RNA quality is expressed as RIN (solid line, error bars are the standard deviation) of 9311-4x (dark color) and 9311-2x (light color). Germination rate is presented as broken lines for 9311-2x (light color) and 9311-4x (dark color). Values were the mean ± SE.

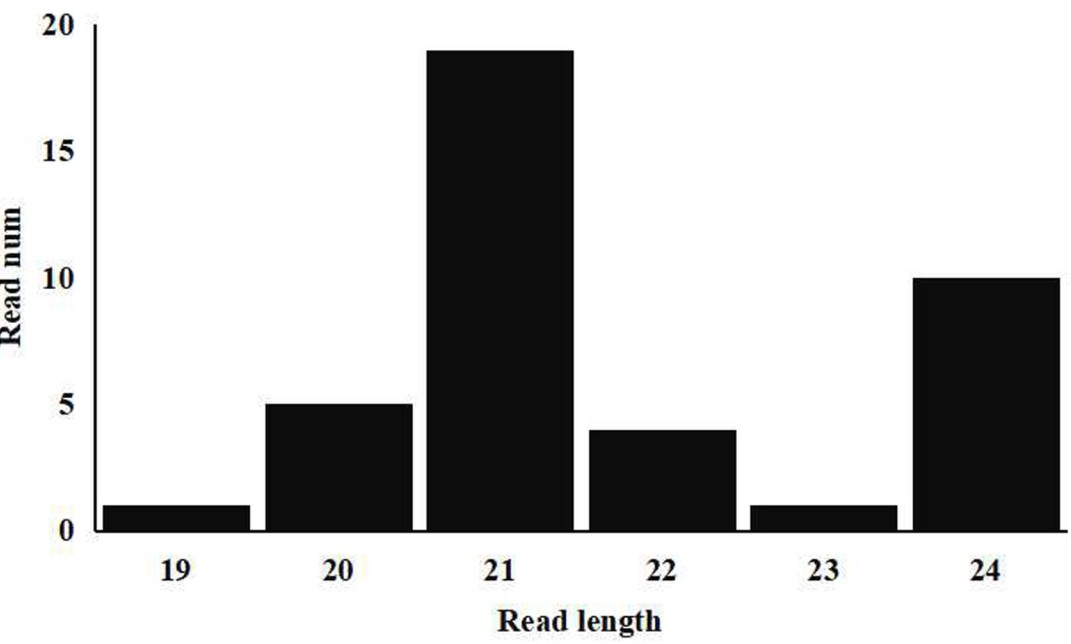

**Fig 4. Conserved miRNA mature length distribution.**

**Table 1. Significantly differently expressed miRNAs in rice seed of 9311-4x *versus* 9311-2x.**

| miRNA_ID | FDR[a]_value | Real FC[b] |
|---|---|---|
| gi_996703422_ref_NC_029266.1__3523 | 0.0000008038 | 4.728999641 |
| gi_996703422_ref_NC_029266.1__3595 | 0.0000008038 | 4.728999641 |
| gi_996703422_ref_NC_029266.1__3794 | 0.0091749584 | 4.683065799 |
| gi_996703425_ref_NC_029263.1__42715 | 0.0003916623 | 3.279495851 |
| osa-miR1846a-3p | 0.0000000000 | 107.4091294 |
| osa-miR1846a-5p | 0.0000043374 | 4.310692117 |
| osa-miR1862e | 0.0068048035 | 2.830525006 |
| osa-miR398b | 0.0000000006 | 5.838129791 |
| osa-miR528-5p | 0.0012456922 | 3.026913983 |
| gi_996703424_ref_NC_029264.1__46071 | 0.0121510891 | 0.239273938 |
| osa-miR164d | 0.0000007643 | 0.222099337 |
| osa-miR1861o | 0.0278668798 | 0.333805444 |

[a]FDR, false discovery rate $<0.05$.

[b]FC, $\log_2$FC (9311-4x expression/9311-2x expression for a gene)$>1$ or $\log_2$FC$<-1$.

Os04g0460600. The interaction analysis shows that gene Os12g0610600 (OsJ_36832) interact with gene Os04g0573200 (CCS), copper chaperone for superoxide dismutase, chloroplastic-like [Oryza sativa Japonica Group (Japanese rice)]. Further analysis shows that gene Os04g0573200 (CCS) interact with gene Os03g0219200 (OS03T0219200-01), Os06g0143000 (OS06T0143000-01), Os06g0115400 (OsJ_19901), Os05g0323900(SODA), Os03g0351500 (SODCC1), Os07g0665200 (SODCC2), OS08g0561700(SODCP). These genes are involved in the regulation of antioxidant enzymes. According to the above analysis, it is suggested that miR164d interacts with antioxidant enzyme genes through its target gene Os12g0610600 (OsJ_36832).

## Target prediction of miRNAs and validation by degradome sequencing

Two degradome libraries were constructed with a balanced mix of RNAs from the three biological replicate samples of each accession. The degradome library sequencing yielded 22,150,505 and 20,490,420 raw reads in 9311-2x and 9311-4x, respectively. After removing the low quality reads, 5′ and 3′ adapter contaminants, and reads smaller than 18 nt, 22,089,602 and 20,429,819 clean reads were obtained, consisting of 5,249,511 and 4,680,655 unique mappable reads for 9311-2x and 9311-4x, respectively. Among these unique mappable reads, 395,259 and 686,180 reads were perfectly matched to the transcripts of rice, with unique mapped ratios of 7.49% and 14.58% for 9311-2x and 9311-4x, respectively (Table 2).

Finally, 38 target genes for 1864 miRNAs were identified, in terms of differential degradation fragments of the transcriptome. Based on the abundance of degradome tags at the target sites, these cleaved targets are listed in S3 Table.

Kyoto Encyclopedia of Genes and Genomes (KEGG) enrichment analysis showed that RNA degradation, peroxisome-associated functions, and plant hormone signal transduction were the most enriched pathways in the two rice accessions (Fig 5). Among them, we found that cleavages of target transcripts of Osa-miR164d were detected with significant differences between diploid and tetraploid accessions. Degraded fragments of miR164d targets were detected by degradome sequencing in tetraploid rice seed, whereas we did not find fragments of miR164 targets in diploid rice seed. Osa-miR164d may be involved in molecular regulation of delayed senescence for rice seed by regulating the differential expression of targeted genes.

**Table 2. Quality evaluation for degradome sequencing data.**

| Sample | 9311-2x (number) | 9311-2x (ratio) | 9311-4x (number) | 9311-4x (ratio) | Total (number) | Total (ratio) |
|---|---|---|---|---|---|---|
| Raw reads | 22150505 | / | 20490420 | / | 42640925 | / |
| Reads < 15 nt after removing 3 adapters | 60903 | 0.27% | 60601 | 0.30% | 121504 | 0.28% |
| Mappable reads | 22089602 | 99.73% | 20429819 | 99.70% | 42519421 | 99.72% |
| Unique raw reads | 5275288 | / | 4704761 | / | 8797333 | / |
| Unique reads<15nt after removing 3 adapters | 25777 | 0.49% | 24106 | 0.51% | 43977 | 0.50% |
| Unique mappable reads | 5249511 | 99.51% | 4680655 | 99.49% | 8753356 | 99.50% |
| Transcript mapped reads | 2579573 | 11.65% | 5143435 | 25.10% | 7723008 | 18.11% |
| Unique transcript mapped reads | 395259 | 7.49% | 686180 | 14.58% | 950717 | 10.81% |
| Number of input transcripts | 43891 | / | 43891 | / | 43891 | / |
| Number of covered transcripts | 17679 | 40.28% | 19843 | 45.21% | 21106 | 48.09% |

## Changes in the transcription of genes related to antioxidant enzymes

Relative expression levels of superoxide dismutase (SOD), catalase (CAT) and peroxidase (POD) genes in seeds of rice accessions 9311-4x and 9311-2x were evaluated by qPCR (Fig 6). The results showed that the relative expression levels of SOD, CAT and POD genes were all up-regulated in 9311-4x, relative to 9311-2x. Os03g0131200 for catalase-1 was significantly upregulated, as were Os06g0547400 for peroxidase-P7, Os03g0343500 for [Cu-Zn] superoxide dismutase and Os06g0143000 for chloroplast-associated [Fe] superoxide dismutase. The expression of these genes was regulated by sRNAs, and the difference in gene expression between 9311-4x and 9311-2x was consistent with the levels of miRNA-164d.

## Integrated miRNA and protein sequencing analysis

In combination with the results of miRNA sequencing and degradome sequencing, we analyzed the relationship between differently expressed miRNAs and the expression of their target genes. Degraded fragments of Osa-miR164d were upregulated in tetraploid rice seeds, which suggested that Osa-miR164d might be downregulated in the tetraploid group. The expression of miR164d was significantly downregulated (log2FC = 5.83) in the tetraploid accession, a finding which was in agreement with the downregulation of Osa-miR164d in this same accession.

In order to determine whether the expression level of different genes ultimately reflected the difference in protein expression, tandem mass tag (TMT) technology was used to identify the protein expression associated with the miRNA targets. The copper/zinc superoxide dismutase 1, copper/zinc superoxide dismutase 2, and catalase isozyme A were all identified as differentially expressed proteins (DEPs) (Table 3). Catalase isozyme A (Protein accession: XP_015625395.1), a member of the catalase family, was the most differentially expressed protein (FC = 1.9) between the two rice accessions. Furthermore, copper/zinc superoxide dismutase 1, and copper/zinc superoxide dismutase 2 were both upregulated in tetraploid rice seeds, although not significantly so, with similar findings for peroxidase P7 (Protein accession: XP_015626941.1) and peroxidase 16 (Protein accession: XP_015640929.1) (log2FC = 1.2).

## Comparison of antioxidant enzyme activities in diploid and tetraploid rice seeds

The enzyme activity of superoxide dismutase (SOD), peroxidase (POD), and catalase (CAT) (Fig 7) was tested in both the tetraploid rice 9311-4x and the conventional diploid rice 9311-2x, the results showing that the activities of these antioxidant enzymes in the seeds of the

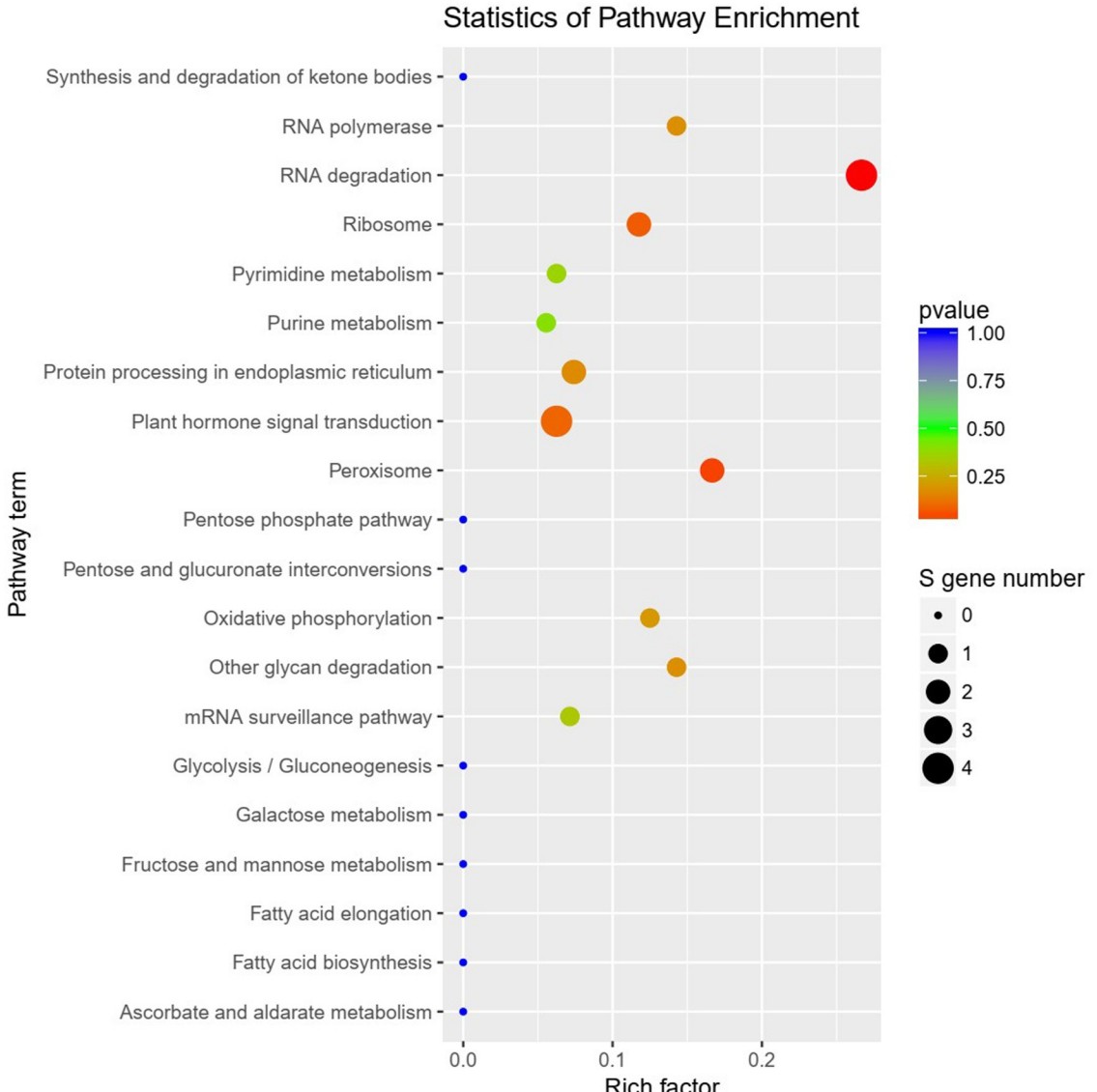

**Fig 5. KEGG pathway enrichment results of different miRNAs.**

tetraploid rice were significantly higher than those of the near-isogenic diploid rice. This trend of antioxidant enzyme activity was consistent with that of the expression of the corresponding antioxidant enzyme regulatory genes and sRNAs. At the same time, the content of malondial-dehyde in different seeds was analyzed (Fig 7). The results showed that the content of malon-dialdehyde, a product of peroxidation of lipids by ROS, was significant higher in diploid rice seeds than that in tetraploid rice seeds, a response which was negatively correlated (r = -0.8927) with the activity of antioxidant enzymes. Therefore, it appears that the increase in antioxidant enzyme activity in tetraploid rice played a positive role in improving the senescence tolerance of tetraploid rice seeds.

ABA induces seed dormancy and inhibits seed germination. The expression of gene OsNCED3 (Os03g0645900) was consistent with the change trend of endogenous ABA content. The gene OsABA8ox2 (Os08g0472800) is invovled in ABA catabolic pathways. The relative

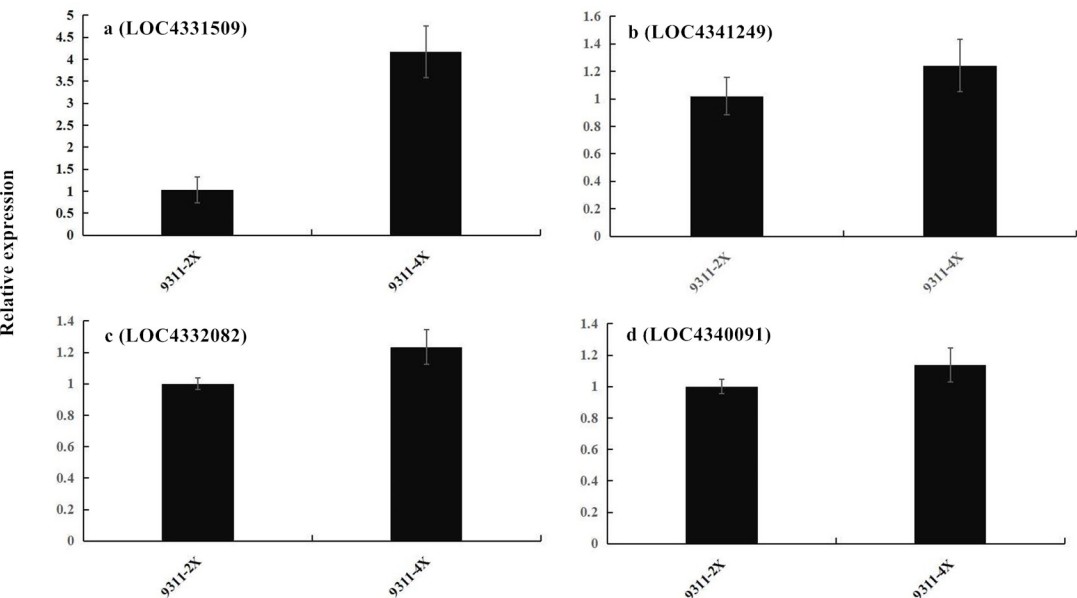

**Fig 6. Changes in the transcription of genes related to antioxidant enzymes.** (a) Os03g0131200 for CAT, (b) Os06g0547400 for POD, (c) Os03g0343500 and (d) Os06g0143000 for SOD, in rice seed 9311-4x and 9311-2x. Values are presented as means ± SE. Significance levels for comparing 9311-4x vs. 9311-2x: Fig 6A: $P < 0.01$; Fig 6B–6D: $P < 0.05$.

expression of genes OsNCED3 and OxABA80x2 (Fig 8) were checked, with the result showing that the 9311-4x rice seeds possessed higher abscisic acid (ABA) synthesis potential and weaker ABA degradation potential than did 9311-2x. As a result, 9311–4 may be in deep dormancy state. Quantitative PCR experiments showed that expression of four genes involved in the synthesis of gibberellins, *OsCPS1*, *OsKO2*, *OsKS1* and *OsGA20*, was upregulated after chromosome doubling treatment, which indicated that the tetraploid rice seeds had greater germination potential than the corresponding diploid.

## Discussion

### Effect of artificial aging on rice seed and RNA integrity

Seed vigor is determined during the dehydration stage of seed development, and the accumulation of nutrient storage material is the basis of seed vigor formation [36, 37]. As the seed matures, the germination rate and vigor of the seed gradually increase and reach a peak at the physiological maturity stage. At this time, both the germination rate and vigor of the seed are at their highest. Seed vigor is affected by genetics, environmental conditions during seed development [38, 39] and storage conditions [40]. In the current study, the use of artificial aging and omics analysis revealed that the tetraploid rice seeds had delayed senescence ability than did seeds of the corresponding diploid accession. From germination data and omics data, the tetraploid accession exhibited greater seed vigor as the senescence period increased, suggesting that the greater senescence tolerance may be caused by key gene regulation.

Seed senescence is a complex biological progress, involving both quantitative and qualitative changes [41]. After harvesting, the physiological and biochemical status of the seed changes with the storage duration, including changes in membrane structure and function,

**Table 3. Functional classifications of identified antioxidant enzyme proteins with significantly altered expression in rice seed 9311-4x *versus* 9311-2x.**

| Protein accession | Protein description | Ratio[a] | P value | MW[b] [kDa] | Cov.[c] [%] | No.p[d] |
|---|---|---|---|---|---|---|
| XP_015625395.1 | catalase isozyme A [*Oryza sativa japonica* Group] | 1.914 | 0.00000314 | 56.697 | 26 | 9 |
| XP_015629749.1 | catalase-1 [Oryza sativa Japonica Group] | 1.093 | 0.13363700 | 56.763 | 15.9 | 5 |
| XP_015643077.1 | catalase isozyme B [Oryza sativa Japonica Group] | 0.973 | 0.23392000 | 56.586 | 31.7 | 9 |
| XP_015615485.1 | probable glutathione peroxidase 2 [Oryza sativa Japonica Group] | 1.346 | 0.00073516 | 22.935 | 15.1 | 3 |
| XP_015622358.1 | peroxidase 24 [Oryza sativa Japonica Group] | 0.773 | 0.00041692 | 33.928 | 8.3 | 2 |
| XP_015622746.1 | probable phospholipid hydroperoxide glutathione peroxidase [Oryza sativa Japonica Group] | 0.958 | 0.04789600 | 25.838 | 39.5 | 6 |
| XP_015626941.1 | peroxidase P7 [Oryza sativa Japonica Group] | 1.184 | 0.03456000 | 34.61 | 4.1 | 1 |
| XP_015628314.1 | peroxidase 35 [Oryza sativa Japonica Group] | 0.884 | 0.00427820 | 33.956 | 20.4 | 6 |
| XP_015630498.1 | L-ascorbate peroxidase 1, cytosolic [Oryza sativa Japonica Group] | 0.939 | 0.05041800 | 27.155 | 41.2 | 7 |
| XP_015632791.1 | probable glutathione peroxidase 4 [Oryza sativa Japonica Group] | / | / | 19.234 | 21.3 | 3 |
| XP_015635542.1 | cationic peroxidase SPC4 isoform X2 [Oryza sativa Japonica Group] | / | / | 36.503 | 7.3 | 2 |
| XP_015635863.1 | probable L-ascorbate peroxidase 7, chloroplastic [Oryza sativa Japonica Group] | 0.987 | 0.72818000 | 38.325 | 18.7 | 4 |
| XP_015635940.1 | probable phospholipid hydroperoxide glutathione peroxidase [Oryza sativa Japonica Group] | 0.727 | 0.00011714 | 18.483 | 44.6 | 6 |
| XP_015637252.1 | cationic peroxidase SPC4 [Oryza sativa Japonica Group] | 0.81 | 0.02661900 | 36.032 | 3.3 | 1 |
| XP_015640929.1 | peroxidase 16 [Oryza sativa Japonica Group] | 1.173 | 0.00049573 | 36.065 | 10.1 | 2 |
| XP_015646556.1 | L-ascorbate peroxidase 2, cytosolic [Oryza sativa Japonica Group] | 0.826 | 0.00207980 | 27.117 | 21.1 | 3 |
| XP_015650808.1 | probable L-ascorbate peroxidase 4, peroxisomal [Oryza sativa Japonica Group] | 0.931 | 0.01002090 | 31.738 | 34.7 | 10 |
| XP_015632609.1 | superoxide dismutase [Cu-Zn] 1 [Oryza sativa Japonica Group] | 0.745 | 0.34 | 15.251 | 53.3 | 3 |
| XP_015640127.1 | superoxide dismutase [Mn], mitochondrial [Oryza sativa Japonica Group] | 1.112 | 0.01 | 24.997 | 48.5 | 8 |
| XP_015647771.1 | superoxide dismutase [Cu-Zn] 2 [Oryza sativa Japonica Group] | 0.896 | 0.05 | 15.081 | 46.7 | 2 |

[a]Ratio between intensities of differentially expressed antioxidant enzyme protein in rice seed 9311-4x *versus* 9311-2x.

[b]Molecular mass of the proteins.

[c]Percentage of the protein sequence covered by matched peptides.

[d]Number of matched unique peptides identified for each protein.

respiration, enzyme activity, synthesis capacity, storage material accumulation, levels of endogenous hormones, etc. [42]. The relative expression of the abscisic acid regulatory genes *OsNCED3* and *OsABA80x2* (Fig 8) in the 9311-4x rice seeds correlated with a greater abscisic acid synthesis ability and a weaker ABA degradation ability, which led to rice seeds entering a deep dormancy state, which could be broken by temperature stimulation. The *OsNCED3* gene plays an important role in synthesis of ABA and in drought stress tolerance in rice [43–45], whereas *OsABA80x2* plays an important role in the degradation of ABA [43]. Quantitative PCR experiments of the expression of *OsCPS1*, *OsKO2*, *OsKS1* and *OsGA20* genes in the three stages of gibberellin synthesis [46–49] showed that the expression of germination stimulatory genes was upregulated in the tetraploid rice, reflecting its higher germination potential.

## Identification of senescence-related miRNAs and their targets by high-throughput sequencing

The results showed that, with the increased duration of senescence time, the integrity of seed RNA decreased gradually, a parameter which was positively correlated with seed vigor. Therefore, it is of great significance to establish a rapid evaluation system of seed vigor based on the RIN value, by comparing and analyzing the differences in RNA integrity between tetraploid

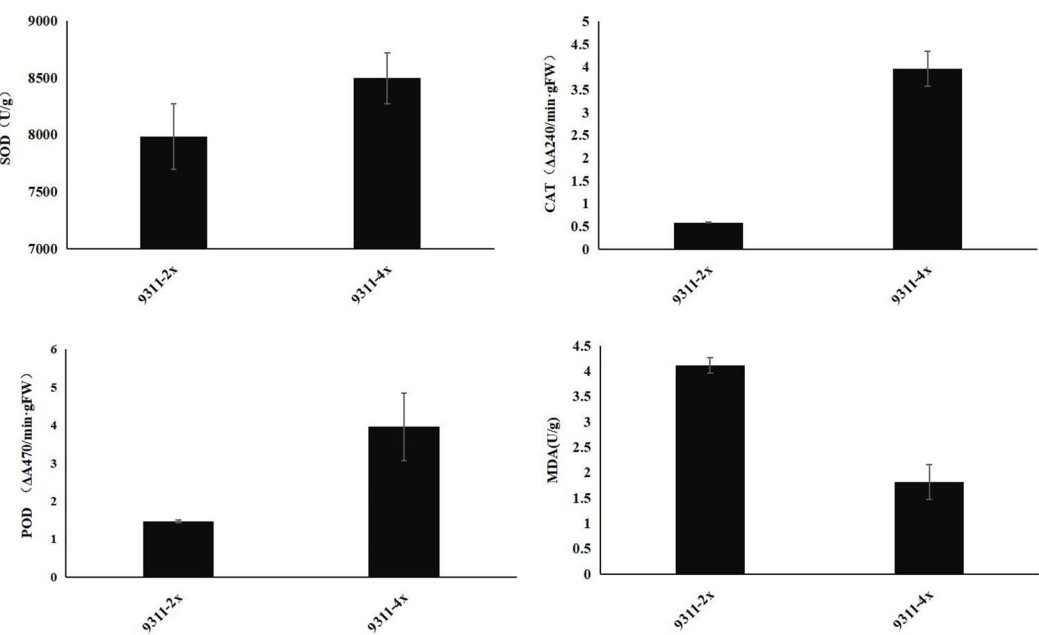

**Fig 7. Comparison of antioxidant enzyme activities and malondialdehyde (MDA) content between rice seeds of 9311-2x and 9311-4x.** (a) The enzyme activity of SOD; (b) the enzyme activity of CAT; (c) the enzyme activity of POD; (d) the content of malondialdehyde. Values are the mean ± SE.

and diploid rice seeds. This present study aimed to clarify the function of the specific miRNA and its targets during seed senescence between diploid and tetraploid rice. Finally, 40 known miRNAs and 58 novel miRNAs were identified in diploid and tetraploid. Only a few rice miRNA targets were identified experimentally, with the majority of miRNA targets being predicted using bioinformatics [50]. Degradome sequencing technology provides a powerful tool with which to study the miRNA-target interactions at the transcriptome level [51]. A total of 38 miRNA was identified by degradome sequencing. Gene function annotation of conserved miRNA targets showed that most of them were classified as transcription factors. Comparisons of the expression levels of miRNAs in the diploid and tetraploid libraries revealed that 12 miRNAs, with a total of 332 target genes, changed significantly. Enrichment results showed that these miRNAs, such as Osa-miR164d, had the functions of hormone signal transduction, peroxisome-associated proteins, and biosynthesis of secondary metabolites, and were significantly more highly regulated in tetraploid rice than in diploid rice. A previous study had revealed that the target of miR164 are members of NAC transcription factor gene family. miR164 was involved in strawberry fruit senescence by negatively mediating the expression of NAC transcription factors [52]. miR164-targeted NAC genes may be negative regulators of drought tolerance in rice [53]. Activation of the stress responses and antioxidant system through downregulating the expression of miR164, miR6260, miR5929, miR6214, miR3946 and miR3446 [54]. In this paper, miR164d targeted NAC gene interacts with antioxidant enzyme genes, including copper chaperone for superoxide dismutase. It is suggested that miR164d may negatively regulate antioxidant enzyme genes, thus delaying the senescence of rice seeds 9311-4x. In our study, degradome libraries were constructed to identify miRNA targets, and transcriptome sequencing was applied to confirm the regulatory relationship between miRNA and

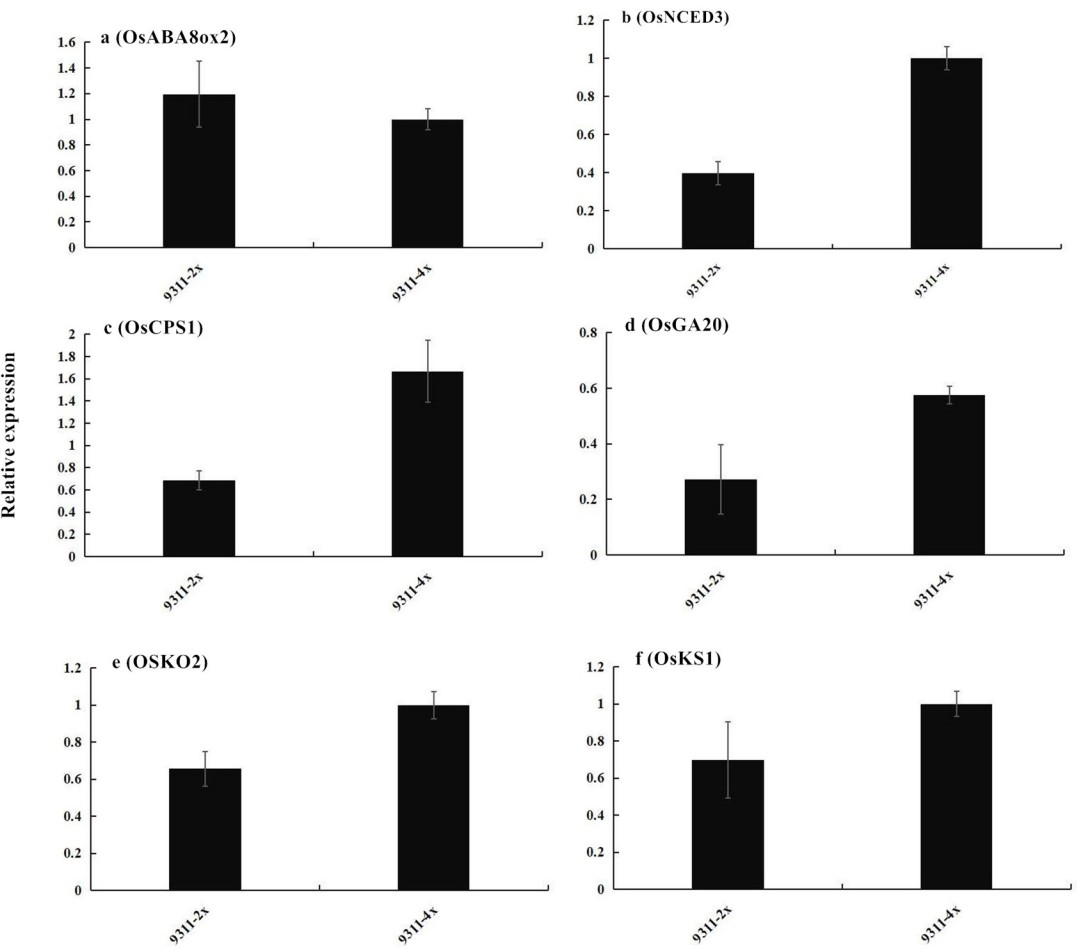

**Fig 8. The relative expression of genes associated with the regulation of abscisic acid (ABA) synthesis/decomposition and gibberellin (GA) synthesis in 9311-2x and 9311-4x rice.** (a) *OsABA8ox2*, gene regulating ABA degradation; (b) *OsNCED3*, gene for ABA synthesis; (c) *OsCPS1*, gene for GA synthesis; (d) *OsGA20*, gene for GA synthesis; (e) *OsKO2*, gene for GA synthesis; (f) *OsKS1*, gene for GA synthesis. Values are the means ± SE.

mRNA. Our results showed that miR164d was expressed at a lower level in tetraploid rice seeds than in diploid rice seeds, a finding which was consistent with our results that tetraploid rice seeds were more capable of withstanding senescence than diploid seeds.

## Integrated analysis of miRNA, degradome sequencing, and protein sequencing

Degradome sequencing was performed to confirm the identity of miRNA targets, and proteome sequencing was used to identify the regulated pattern of miRNA targets. With a combination of multiple omics, a series of enzymes with antioxidant activity, targeted by Osa-miR164d, were quantified. During senescence, peroxidases create a non-toxic environment by the massive production of ROS to prevent cellular diffusion of superoxide. Peroxidase P7 and peroxidase 16 were identified as up-regulated proteins in tetraploid rice seeds. Peroxidase P7 functioned by removal of $H_2O_2$, oxidation of toxic reductants, biosynthesis and degradation of

lignin, suberization, auxin catabolism, and response to environmental stresses, such as wounding, pathogen attack, and oxidative stress [55–57]. Catalase isozyme A exhibited a marked increase in expression of various classes of CAT proteins. The fold change value of catalase isozyme A was significantly higher than those of other catalases and catalase isozymes. The upregulation of catalase isozyme A in tetraploid rice can protect seeds from hyperoxidation. The quantitative proteomics results were consistent with those from enzyme assays, which indicating that the quantitative proteome data were reliable.

The data indicated that antioxidant enzymes played a positive role in improving the tolerance to senescence of tetraploid rice seeds. In the process of long-term seed storage, seed senescence, affected by seed chemical compound composition, storage conditions, and other factors, will cause seed deterioration, resulting in a decline in seed vigor, accompanied by a series of physiological and biochemical changes. It is generally believed that the cause of seed deterioration is lipid peroxidation, caused by free radicals [58]. Free radicals and ROS have strong oxidative effects. Some free radicals are free in the cell, leading to free radical chain reactions, which lead to the peroxidation of unsaturated fatty acids on the membranes, causing the destruction of the structure of the biological membrane, and a decrease in the content of protein, nucleic acid, and other biological macromolecules, leading to seed senescence and death. In the process of seed senescence, when the balance between oxidants and antioxidants is destroyed, the excess free radicals will lead to or aggravate the membrane lipid peroxidation, and volatile acids, such as malonic acid, will cause damage to the cell membrane system, causing the metabolism disorder, and resulting in the rapid decline in seed vigor and even seed death. Malondialdehyde is the product of membrane lipid peroxidation, which is the direct expression of the degree of membrane peroxidation. The lower the content of MDA, the lower the degree of membrane damage; conversely, seed vigor was positively correlated with antioxidant enzyme activity, but was negatively correlated with MDA content [59].

In summary, the present study is the first attempt to integrate miRNA and protein expression, along with degradome analysis, to identify key regulatory miRNA-targets in rice during the senescence. Data indicated that antioxidant enzyme activity may play an important role in tolerating senescence by the regulation of miRNA-164d.

## Conclusions

Seed senescence tolerance is of great importance to seed quality and grain production. Taking tetraploid rice as the experimental material, the findings of the current study will provide a new vision for the research into delayed senescence in rice seeds. To develop a better understanding of the molecular level variations among diploid and tetraploid rice seeds, we sequenced the normal diploid seeds (9311-2x) and their near-isogenic tetraploid counterpart seeds (9311-4x). The miRNA analysis led to the discovery of internal regulatory mechanisms of transcript degradation between the diploid and tetraploid variants. Some miRNA families and their targets, related to seed senescence, were identified by degradome sequencing and proteomics sequencing. These results provide useful information for the study of miRNAs in seed senescence miRNAs in rice. Further studies are required to identify these miRNAs and their targets, using experimental strategies.

## Supporting information

**S1 Table. List of conserved miRNA and novel miRNA.**
(XLS)

**S2 Table. Precursor of converted miRNA and novel miRNA.**
(XLS)

**S3 Table. Degradome result.**
(XLS)

## Acknowledgments

We thank International Science Editing (http://www.internationalscienceediting.com) for editing this manuscript.

## Author Contributions

**Writing – original draft:** Baosheng Huang, Lu Gan, Dongjie Chen, Yachun Zhang, Yujie Zhang, Xiangli Liu, Si Chen, Zhisong Wei, Liqi Tong, Zhaojian Song, Xianhua Zhang, Detian Cai.

**Writing – review & editing:** Changfeng Zhang, Yuchi He.

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
