## [Decision Letter · Decision Letter 0]

11 Sep 2020

PONE-D-20-22960

Integration of Small RNA, Degradome and Proteome Sequencing in Oryza sativa Reveals a Delayed Senescence Network in Tetraploid Rice Seed

PLOS ONE

Dear Dr. He,

Thank you for submitting your manuscript to PLOS ONE. After careful consideration, we feel that it has merit but does not fully meet PLOS ONE’s publication criteria as it currently stands. Therefore, we invite you to submit a revised version of the manuscript that addresses the points raised during the review process.

We look forward to receiving your revised manuscript.

Kind regards,

Baohong Zhang, Ph.D

Academic Editor

PLOS ONE

Journal Requirements:

2.We suggest you thoroughly copyedit your manuscript for language usage, spelling, and grammar. If you do not know anyone who can help you do this, you may wish to consider employing a professional scientific editing service.  

[This project was supported by the Chinese National Natural Science Foundation (Grant Nos.31960068，31270356, 31271690, and 31571639),  2017 Hubei Science and Technology Department Innovation Team (2017CFA023), 2016 Wuhan Yellow Crane Talents (science) Foundation.].   

We note that one or more of the authors are employed by a commercial company: Wuhan Polyploid Biology Technology Co. Ltd,

Reviewers' comments:

Reviewer's Responses to Questions

**Comments to the Author**

1. Is the manuscript technically sound, and do the data support the conclusions?

Reviewer #1: Yes

Reviewer #2: No

2. Has the statistical analysis been performed appropriately and rigorously? 

Reviewer #1: Yes

Reviewer #2: No

3. Have the authors made all data underlying the findings in their manuscript fully available?

Reviewer #1: Yes

Reviewer #2: No

4. Is the manuscript presented in an intelligible fashion and written in standard English?

Reviewer #1: Yes

Reviewer #2: No

5. Review Comments to the Author

Reviewer #1: The manuscript describes a technically sound scientific study with data to support the findings. Experiments were performed with appropriate controls, repetitions and sample sizes. Conclusions are duly drawn from the data presented. The statistical analysis been performed appropriately. The authors have made fully available all the data underlying the conclusions in their manuscript. The manuscript presented in an intelligible fashion and written in standard English

Reviewer #2: This manuscript described a comprehensive comparison between tetraploid and diploid rice accessions via miRNA-seq together with Degradome and proteome sequencing analyses. The authors provide preliminary data showing that miR164d and some other small RNAs may play roles in regulation of the activity of antioxidant enzymes, leading to determining of seed senescence and seed longevity. However, I did not found any degradome and proteome data to support the claims throughout the manuscript. Besides, I have the following concerns to argue the publishable of this manuscript.

There are many confusing or unclear sentences, I listed some below.

Lines 42-64, the two sentences here are confusing.

Lines 84-90, the long sentence here is confusing.

Lines 97-98, the mean of the sentence “With…decline” is unclear.

Line 121, “identifying” is not a correct word here, how can “mechanisms” be identified? I suggest to use “understanding” instead.

Lines 343-344 and Fig 6, the gene IDs for SOD, CAT and POD are not clear. I suggest that the authors use either RAP locus (for example, Os02g0672200) or MSU locus (LOC_Os02g45070), so that the readers can see what genes are been talking about.

Lines 364-368 and Table 3, I cannot understand what the protein accession numbers are used. Are the numbers from the database of LC-MS/MS or from any public databases? If these numbers are from the database of LC-MS/MS, please change to numbers corresponding to those of a public database. If the numbers are from a public database, please point out what database is used.

Other concerns:

Lines 53-54 and throughout the manuscript, delete the “-” between “miR” and the “number”. The name of miRNAs should follow the previous reports.

Lines 103-111, I suggest to move “MicroRNAs…process” into a new paragraph.

Lines 110-111, the statement of the sentence “Previous studies…” seems not consistent with the context and the cited reference (33) is incorrect.

Lines 118-119, I cannot understand how proteomics can be used to verify the miRNA and degradome sequencing results. RNA sequencing data should be verified by RT-PCR.

Lines 70-74 and 125-128, the authors reasoned that PMeS rice lines have advantages, I am suspicious that the polyploidy 9311-4x is with high seed setting. I suggest that the authors show the panicle images to display this trait.

Line 136, “6000lx” should be “6000 lx”.

Line 137, “20 seeds for” should be “Twenty seeds were used for”

Lines 221-223, antioxidant enzyme activity assays should cite references or give a brief description of the method.

Line 291, here should cite previous published papers because of the claim “consistent with previous studies in O. sativa”.

Lines 305-306 and table 1, I do not think that miR164d targets a gene encoding Cu-Zn superoxide. I suggest that the authors double check and cite a reference for this claim. I also suggest that the authors double check the fold change (FC) value in table 1 and clarify whether they are log2FC.

Line 326, there is not Supplement Table S3.

Lines 329-332 and 356-358, I did not find the data in Fig 5 or any other place to support the claims here. I suggest that the authors present the degradome data in a supplemental data set.

Lines 394-396, the authors should be aware that the expression of OsNCED3 and OxABA80x2 is far from ABA biosynthesis or degradation. I suggest that the authors revision the claims here. Besides, what is OxABA80x2? Please give the gene ID.

Lines 448-456, superoxide dismutase genes are not the target of miR164, instead, they are the target of miR398. The target of miR164 are members of NAC transcription factor gene family.

6. PLOS authors have the option to publish the peer review history of their article (what does this mean?). If published, this will include your full peer review and any attached files.

Reviewer #1: **Yes: **Professor Anatoliy Ivashchenko

Reviewer #2: **Yes: **Wen-Ming Wang

---

## [Author Response · Author response to Decision Letter 0]

27 Oct 2020

Dear editor,

We received the reviews from two reviewers on September 11, 2020. First of all, we would like to thank you and the two reviewers very much for giving us valuable suggestions about revising our paper (PONE-D-20-22960). We have earnestly revised this paper according to the comments made by you and the two reviewers. The specific revisions are highlighted in “Revised Manuscript with Track Changes” and the answers are shown in “Response to Reviewers”. All figure files have been processed by the Preflight Analysis and Conversion Engine (PACE) digital diagnostic tool. Financial Disclosure and Conflicts of Interest statements have been updated in cover letter. We look forward to your valuable suggestion.

Kindest regards,

Yuchi He, Changfeng Zhang

---

## [Editor Report · Decision Letter 1]

30 Oct 2020

Integration of Small RNA, Degradome and Proteome Sequencing in Oryza sativa Reveals a Delayed Senescence Network in Tetraploid Rice Seed

PONE-D-20-22960R1

Dear Dr. He,

We’re pleased to inform you that your manuscript has been judged scientifically suitable for publication and will be formally accepted for publication once it meets all outstanding technical requirements.

Kind regards,

Baohong Zhang, Ph.D

Academic Editor

PLOS ONE
---

## [Editor Report · Acceptance letter]

5 Nov 2020

PONE-D-20-22960R1 

Integration of Small RNA, Degradome and Proteome Sequencing in *Oryza sativa* Reveals a Delayed Senescence Network in Tetraploid Rice Seed 

Dear Dr. He:

I'm pleased to inform you that your manuscript has been deemed suitable for publication in PLOS ONE. Congratulations! Your manuscript is now with our production department. 

Kind regards, 

on behalf of

Professor Baohong Zhang 

Academic Editor

PLOS ONE